# Longitudinal ridges imparted by high-speed granular flow mechanisms in martian landslides

Giulia Magnarini [1]*, Thomas M. Mitchell [1], Peter M. Grindrod [2], Liran Goren [3] & Harrison H. Schmitt[4]

The presence of longitudinal ridges documented in long runout landslides across our solar system is commonly associated with the existence of a basal layer of ice. However, their development, the link between their occurrence and the emplacement mechanisms of long runout landslides, and the necessity of a basal ice layer remain poorly understood. Here, we analyse the morphometry of longitudinal ridges of a martian landslide and show that the wavelength of the ridges is 2–3 times the average thickness of the landslide deposit, a unique scaling relationship previously reported in ice-free rapid granular flow experiments. We recognize en-echelon features that we interpret as kinematic indicators, congruent with experimentally-measured transverse velocity gradient. We suggest that longitudinal ridges should not be considered as unequivocal evidence for presence of ice, rather as inevitable features of rapid granular sliding material, that originate from a mechanical instability once a kinematic threshold is surpassed.

[1] Department of Earth Sciences, University College London, London, UK. [2] Natural History Museum, London, UK. [3] Department of Geological and Environmental Sciences, Ben Gurion University of Negev, Beer-Sheva, Israel. [4] Department of Engineering Physics, University of Wisconsin Madison, Madison, WI, USA. *email: giulia.magnarini.14@ucl.ac.uk

D istinctive longitudinal raised ridges are common in large-scale mass movements across the surfaces of planetary bodies, but their formation mechanism is poorly understood. These longitudinal ridges have been identified in impact ejecta on every planetary surface with sufficient image resolution[1], but are best documented in long-runout landslides[2–4]. The emplacement dynamics of long-runout landslides is a subject of ongoing debate[2,5–9], and various mechanisms have been proposed for the reduction of basal friction, which appears to be a condition necessary for controlling the ability of long-runout landslides to travel even more than ten times their vertical drop over nearly flat surfaces[2,5,8,10].

Longitudinal ridges and furrows are seen ubiquitously on the surface of long-runout landslides on Mars. On Earth, such features are commonly found in landslides emplaced on glaciers, amongst which is the iconic Sherman Glacier landslide[11]. Consequently, their morphological similarity to those observed in martian landslides has been used as an analogy to infer the presence of an icy substrate at the time of landslide emplacement[12,13] on Mars. It has been proposed that presence of ice on the surface reduces the frictional coefficient, permitting the splitting of the deposit as a consequence of either lateral spreading[13] or the dominant role of inertia over lateral spreading forces[12], leading to the formation of ridges and troughs perpendicular to the spreading direction. The presence of ice is also suggested to explain the development of longitudinal grooves in martian double-layered ejecta craters[14], which also show morphological similarity to the longitudinal ridges associated with landslides, although other mechanisms are not excluded[1,15]. However, longitudinal ridges have been observed also in landslides on the Moon[16–18], which is considered to be free of ice throughout its geological history.

Borzsonyi et al.[19] suggested the possibility that longitudinal ridges and furrows of large planetary rock avalanches may be related to fluid-like instabilities forming convection cells that result in longitudinal stripes parallel to the flow direction. This was based on previous experimental and theoretical work[20,21] demonstrating turbulent flow processes that are well known to occur in fluid mechanics. The experimental work and stability analysis on rapid granular flows conducted by Forterre and Pouliquen[20,21] showed for the first time the spontaneous formation of longitudinal vortices that manifest as superficial longitudinal ridges and furrows, which initiation depends on the velocity of the flow, roughness of the basal surface, and the consequent strong shear at the base of the flow. Importantly, Forterre and Pouliquen[20] demonstrated a clear morphological scaling relationship in that the wavelength of the ridges scales with the thickness of the flow by a factor of ~2–3 and that the pattern drifts in the transverse direction. Further experiments and numerical modelling by Borzsonyi et al.[19] confirmed that the longitudinal pattern naturally arises from mechanical instabilities within rapid granular flows. While demonstrated experimentally, such scaling relationship have yet to be demonstrated in any planetary landslide.

Here, we use state-of-the-art Mars Reconnaissance Orbiter imagery (CTX[22] and HiRISE[23] cameras) in order to analyse a giant martian landslide with some of the best defined longitudinal ridges (Fig. 1) for comparison with the longitudinal morphologies obtained during granular experimental slides, so to investigate the possibility that the same instability may be responsible for the longitudinal ridges and furrows of large planetary rock avalanches, as suggested by Borzsonyi et al.[19]. We find similar scaling relationship between the wavelength of longitudinal ridges and the thickness of the deposit as seen in the martian landslide and in experiments of self-forming ice-free ridges. We infer that the emplacement velocity of the martian landslide could have been as high as tens to hundreds of m/s. On the basis of these results, we suggest that longitudinal ridges that characterise long-runout terrestrial and planetary landslides may be the expression of an instability that emerges within the flowing mass, as observed in laboratory experiments on rapid granular flows[20].

## Results

**Ridges morphometry with distance and with deposit thickness**. We identified longitudinal ridges on the surface of the landslide deposit. We chose the two areas with the best exposed ridges to conduct our analysis (Fig. 1c, d) and traced profiles transversal to the flow direction to study how ridges evolve with distance. We measured the spacing between the ridges and calculated their density (number of ridges/profile length) in each profile (see the Methods section). Within both areas, from the proximal to distal edge, we found that ridges diverge while their number increases and their amplitude and wavelength decrease (Fig. 2a). At each profile, we also assessed the average spacing between the ridges (S) and the local average thickness of the deposit (T) and calculated the S/T ratio, as done by Forterre and Pouliquen[20] (Supplementary Tables 1 and 2). The central area (Fig. 1c and blue in Fig. 2) shows an almost constant S/T ratio, between 2.55 and 2.99. The east area (Fig. 1d and yellow in Fig. 2) shows lower values, yet, still with small variability that ranges between 1.44 and 2.4.

Our results suggest a scaling relationship between the wavelength of ridges and the average thickness of the landslide deposit. We compare the S/T ratio obtained from our morphometric analysis with the results obtained by Forterre and Pouliquen[20]. In particular, the values of the S/T ratio obtained in the central area (Fig. 2b) are in agreement with the value of ~2–3 found by Forterre and Pouliquen[21]. In the east area (Fig. 2b), profiles P3, P4 and P5 show S/T ratio below 2, which is a consequence of a localised increase in the deposit thickness (Supplementary Table 2). We observed the appearance of new ridges between diverging ridges (Fig. 3a–d), confirming that the number of ridges increases with distance, as found from the morphometric analysis (Supplementary Tables 2 and 3). This is also consistent with the observations of Forterre and Pouliquen[21] of transversal vortex drifting accompanied by annihilation and creation of ridges. Divergence of ridges is also observed in martian double-layered ejecta craters[1,24].

**Kinematic indicator features and flow structure models**. We also identified s-shaped en-echelon features superposed on longitudinal ridges that we interpret as kinetic indicators (Fig. 3e–h). Laboratory experiments have shown the existence of a velocity gradient between ridges and furrows: for very rapid flows (dilute regime), furrows move faster than ridges[19,20], whereas for less rapid flows (dense regime), the ridges are the faster-flowing region[19]. In both regimes, the faster-flowing regions are denser, i.e., their solid fraction is greater than the slower-flowing features. As a consequence, when the flow ceases, inversion of topography is expected in the dilute regime. Here, the ridges, being less dense, are expected to deflate and sink. The ridges become furrows, and the fast-flowing dense furrows become ridges. The en-echelon features that we report in Fig. 3 pointing at faster-flowing ridges is therefore consistent with both the dense regime and with the dilute regime after topography inversion (Fig. 4).

## Discussion

A self-lubrication mechanism for long-runout landslides has been proposed by Campbell[25] suggesting that a dilute layer of highly

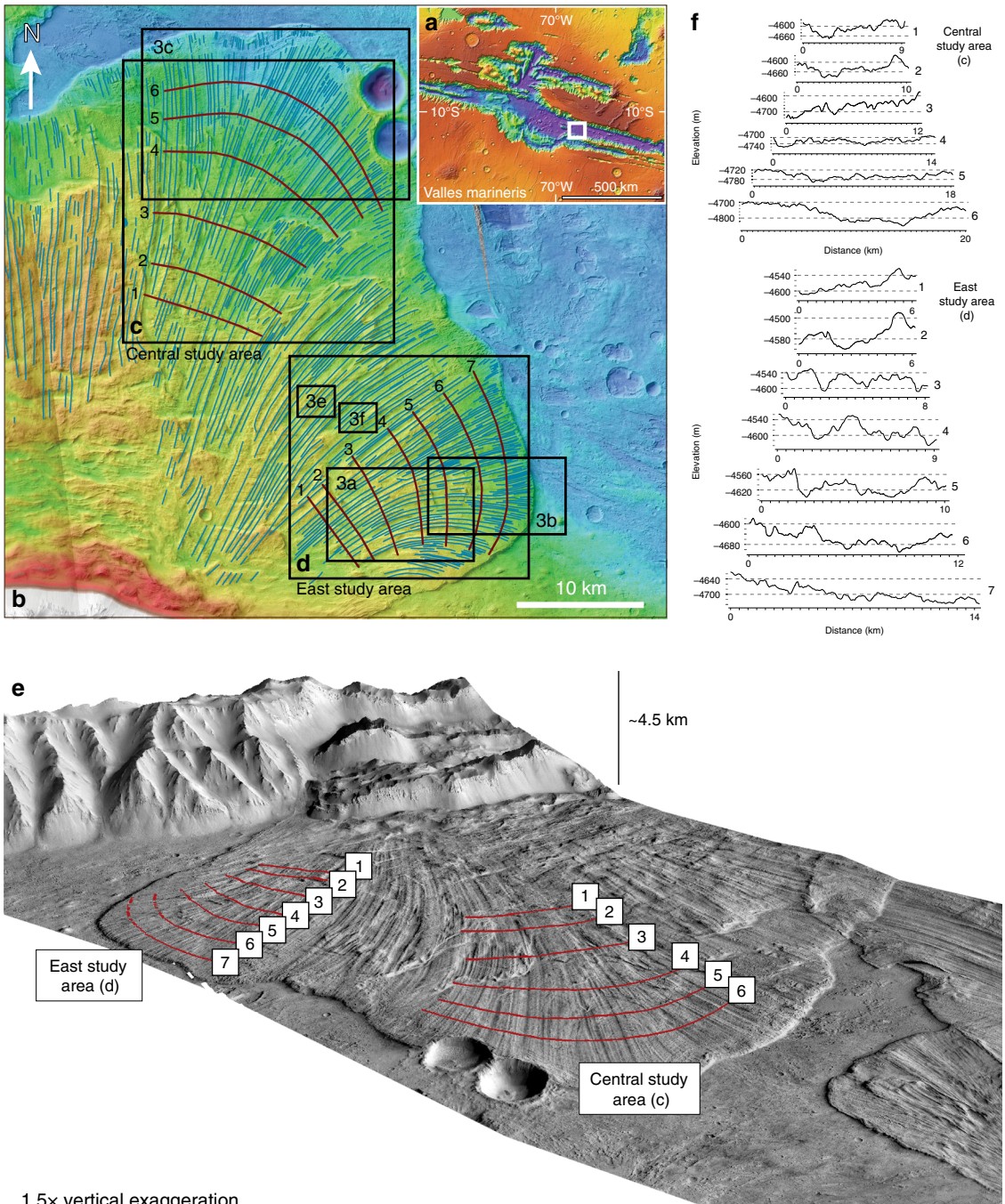

**Fig. 1** Area of the study. **a** Valles Marineris, Mars, showing the location of the landslide that we study here in Coprates Chasma (white box). **b** CTX Stereo-derived DEMs. **c**, **d** Study areas within which morphological and morphometric characterisation is conducted; burgundy lines: topographic profiles, numbered from proximal to distal edge of the landslide deposit; blue lines: longitudinal ridges. (3a), (3b), (3c), (3e), (3f): locations of close-up images shown in Fig. 3. **e** Oblique view of the landslide object of the study. Red lines represent topographic profiles, numbered from proximal to distal area (following flow direction). **f** Topographic profiles of the landslide deposit of the east and central areas of the study

agitated particles at the bottom of the flow can explain the apparent low coefficient of friction in long-runout landslides and that such a layer should form naturally in rapid granular flows. As the formation of longitudinal vortices also seems to be a natural development of rapid granular flows that involves a low-density layer at the bottom of the flow[19–21], it is not unreasonable to suggest that longitudinal ridges and furrows are inevitable features that originate from a mechanical instability once certain threshold conditions within a flow moving over a rough surface are passed.

The mechanism responsible for the formation of longitudinal vortices proposed by Forterre & Pouliquen[20] relies on the inversion of the density profile, with denser granular packing closer to the top of the slide, and on a continuous energy supply to the system, in the form of shear forces, so to maintain the collisional regime and, therefore, to prevent the granular temperature to decay rapidly due to the inelastic nature of particle collisions[25]. The instability derives from the inversion of density profile, due to an increase of granular temperature[26,27] at the base of the flow by collision and shear between particles and the rough

**a**

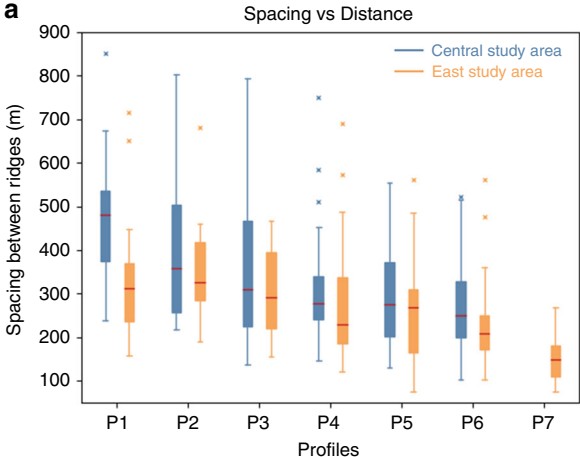

**b**

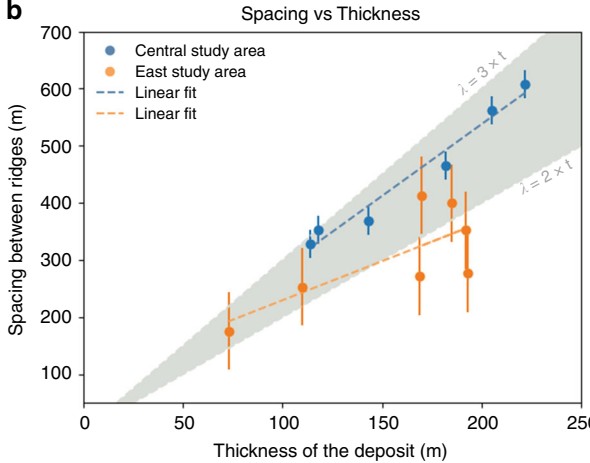

**Fig. 2** Variation of the spacing between longitudinal ridges with the distance and with the thickness of the deposit. Panel **a** shows how the spacing between longitudinal ridges varies with distance (from the proximal to distal areas of the deposit); boxes represent the 'interquartile range', red segments within boxes show the median value, whiskers above and below boxes show the locations of the maximum and minimum, symbols 'x' represent values above three times the interquartile range. Panel **b** shows the relationship between the spacing of the ridges and the thickness of the deposit; the grey band shows the experimentally predicted range for the scaling relationship between the spacing between the ridges and the thickness of the deposit[20]

substrate. Such particle interactions may well contribute to acoustic fluidisation[28] processes within the landslide that would lead to the dynamic reduction of viscosity and explain the long-runout distance. Several other dynamic weakening mechanisms have also been proposed to explain long run-outs, such as flash heating[3], frictional melting[29], and thermal decomposition[30–33] (and subsequent thermal pressurisation). Such mechanisms rely on the generation of heat at the base of the landslide, which would contribute to the energy supply at the base of the system.

From the results of our morphometric analysis of longitudinal ridges in a giant martian landslide appears the similarity of the scaling relationship between the wavelength of longitudinal ridges and the thickness of the landslide seen in nature and in experiments of self-forming ice-free ridges. Moreover, scaling analysis of the martian landslide studied here indicates that its emplacement velocity could have been as high as tens to hundreds of m/s (see the Methods section). It follows that longitudinal ridges that characterise long-runout terrestrial and planetary landslides may

be the expression of an instability that emerges within the flowing mass once a velocity threshold is surpassed, as observed in laboratory experiments on rapid granular flows[20]. Although not ruling out the presence of ice and other mineralogical facies as a key factor, our results suggest that the origin of longitudinal ridges does not necessarily depend on the presence of an icy substrate.

Our results and observations represent the first field evidence of experimental and numerical studies that show the spontaneous development of longitudinal ridges in ice-free rapid granular flows, and support the existence of a convincing fundamental process responsible for the origin of such morphologies, challenging existing analogies to terrestrial landslides on ice that have been proposed to explain their widespread occurrence throughout the Solar System. If indeed the scaling relations that we identify indicate that the martian landslide that we study developed longitudinal ridges through an instability that emerges from the physics of fast-flowing dry grains, then we should expect such morphological features throughout the Solar System, regardless of the availability of other lubrication mechanism, such as ice layers.

## Methods

**Satellite image data and stereo-derived topography data**. Morphological mapping of the martian landslide object of this study (Coprates Chasma, 67.75W-11.80S) was conducted using Mars Reconnaissance Orbiter imagery (CTX[22] and HiRISE[23] cameras) provided by the NASA Planetary Data System. This study has made use of the USGS Integrated Software for Imagers and Spectrometers (ISIS)[34] package to pre-process CTX stereo-pair images (Supplementary Table 3). The processed CTX images were imported in SOCET SET commercial suite from BAE Systems to obtain digital elevation models and ortho-rectified images using well-validated previous methods[34]. These image products were post-processed with ISIS to create final images that could be used in ArcGIS.

**Morphological characterisation methods**. In ArcGIS, longitudinal ridges are traced in correspondence of the crest. Two sets of transversal topographic profiles were traced (Fig. 1c, d) so that the most external ridges (at the left and right end) were always the same ones in every profile. Distances between ridges were measured manually with the ruler tool. The number of ridges and their density along each profile were obtained using a custom Python script (see the Code availability section). In order to assess the average thickness of the landslide deposit in the correspondence of every profile, we built the topography of the valley floor underneath the deposit by interpolating topographic contours of Valles Marineris floor around the landslide (Supplementary Figs. 1 and 2). Because the CTX-derived DEM and the newly built DEM representing the valley floor have different resolution, the number of elevation points available is different, therefore preventing a direct correlation and thickness assessment point by point. We overcame this aspect by taking elevation points of the valley floor and the correspondent closest elevation points (one before and one after) of the landslide deposit. We then averaged the elevation between these two points and calculated the thickness. The thickness values obtained per each point of the valley floor were finally averaged. We automated these steps by means of a custom Python script (see the Code availability section).

**Landslide deposit thickness error estimation**. We traced profiles transversal and longitudinal to the valley floor, adjacent to the central and east areas of the study (Supplementary Fig. 3). This set of transects (yellow lines), in the specific their linear fits (burgundy lines), is compared with a mirror set of transects traced in the correspondence of the landslide deposit (blue lines) so to assess the plausibility of the overall elevations of our reconstructed bottom surface, and to provide an error estimation of the average thickness that we provide in the paper.

As the floor of Valles Marineris gently dips towards east (in the figure, from left to right), the higher elevations of the blue lines 1 and 2 are plausible. However, we are also expecting a random natural roughness of the valley floor at various scales so that also blue lines 3 and 4 appear plausible, as their maximum offset (blue line 3) is ~80 m, which is within a reasonable magnitude of topographic variation (as seen in profile 1, Supplementary Fig. 3).

We use these profiles to evaluate the errors on the deposit thickness, and we do so by accounting for the standard deviation of the elevation in the topographic profiles (Supplementary Table 4). To do so, we use the topographic profiles that we traced in the valley floor, adjacent to the central and east areas of study (yellow lines). As the topographic profiles that we traced for the morphometric analysis are parallel to neither transversal nor longitudinal direction, we determine the final error taking into account the propagation of error along these two axes (Eq. (1)), that is combining standard deviation for yellow lines 1 and 2 to obtain the error in

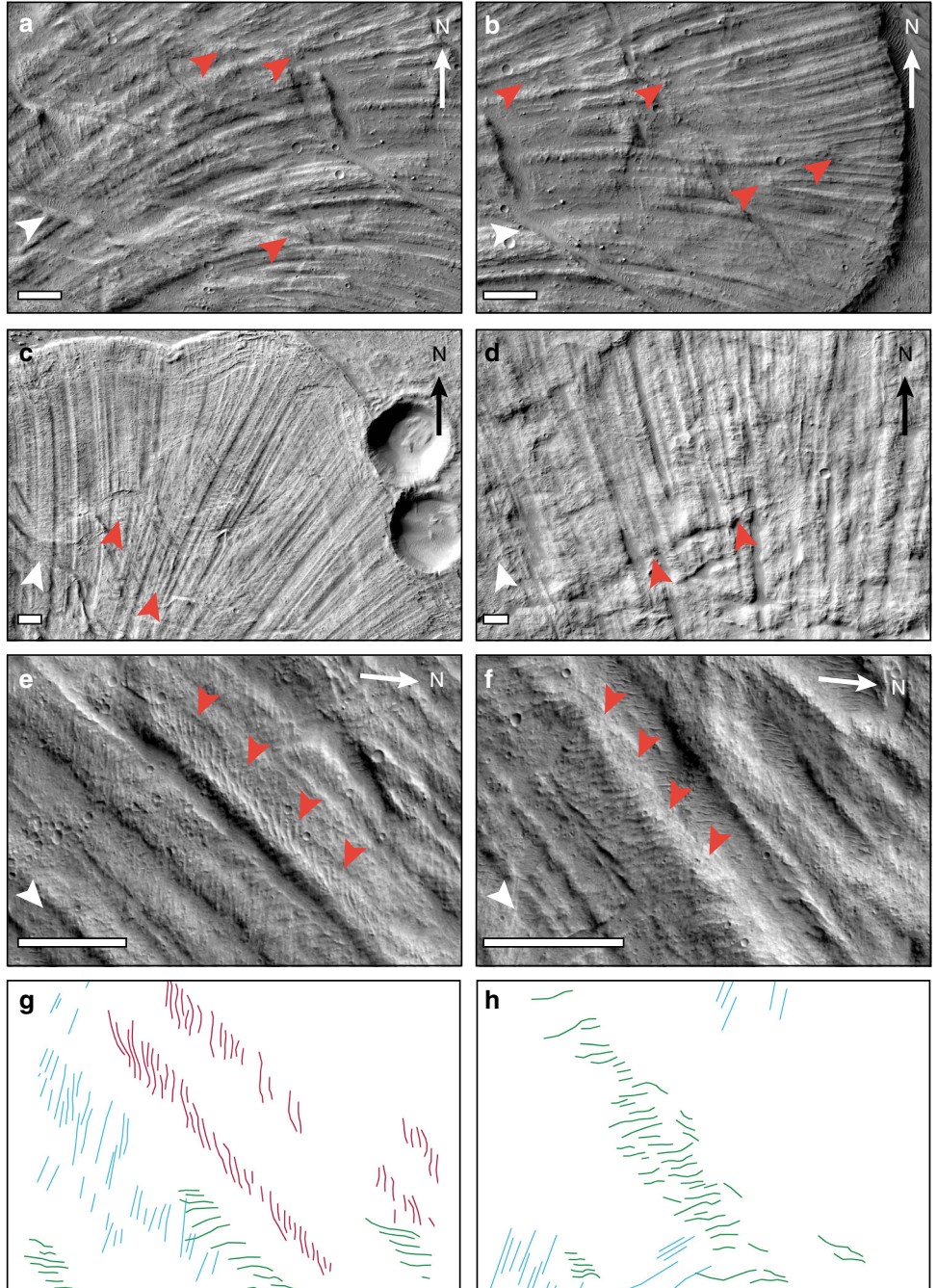

**Fig. 3** Close-up images showing the behaviour of longitudinal ridges and superposed en-echelon features. **a–d** appearance of new, smaller ridges between diverging ridges, as indicated by orange arrowheads; this behaviour may represent field evidence for laboratory observations made by Forterre and Pouliquen[20], who reported of a complex non-linear evolution that includes transversal drift and creation of longitudinal streaks in rapid granular flows. **e**, **f** s-shaped en-echelon features superposed on longitudinal ridges; these features may represent kinematic indicators, providing evidence of a velocity gradient between ridges and furrows. **g**, **h** Digitalised en-echelon features observed on ridges in (**e**) and (**f**), respectively; red and green colour represent s-shaped features with a rough east-west and north–south orientation, respectively; cyan colour represents other linear features that are not considered in this study. Image (3a), (3b) and (3c) locations are shown in Fig. 1b; location of (3d) is westward to Fig. 1b. White arrowheads indicate flow direction. Scale bar is 1 km

the central area of study (c) and yellow lines 3 and 4 for the error in the east area of study (d), represented as error bars in Fig. 2b of the main text:

$$\sigma_k = \sqrt{\sigma_i^2 + \sigma_j^2} \qquad (1)$$

**Velocity estimation**. The experimental work of Börzsönyi et al.[19] identified a range of scaled downstream surface velocity.

The flow velocity $u$ is scaled with $\sqrt{g * d}$, where $g$ is the gravitational acceleration and $d$ is the grain size, providing the normalised downstream surface velocity $\tilde{u}^S$:

$$\tilde{u}^S = \frac{u^S}{\sqrt{g * d}} \qquad (2)$$

From the data in Fig. 3 of Börzsönyi et al.[19], we consider $\tilde{u}^S = 40$, as this is a value for which the longitudinal ridges develop. We use the gravitational acceleration for Mars, 3.71 m/s², and the largest grain sizes recognisable on the

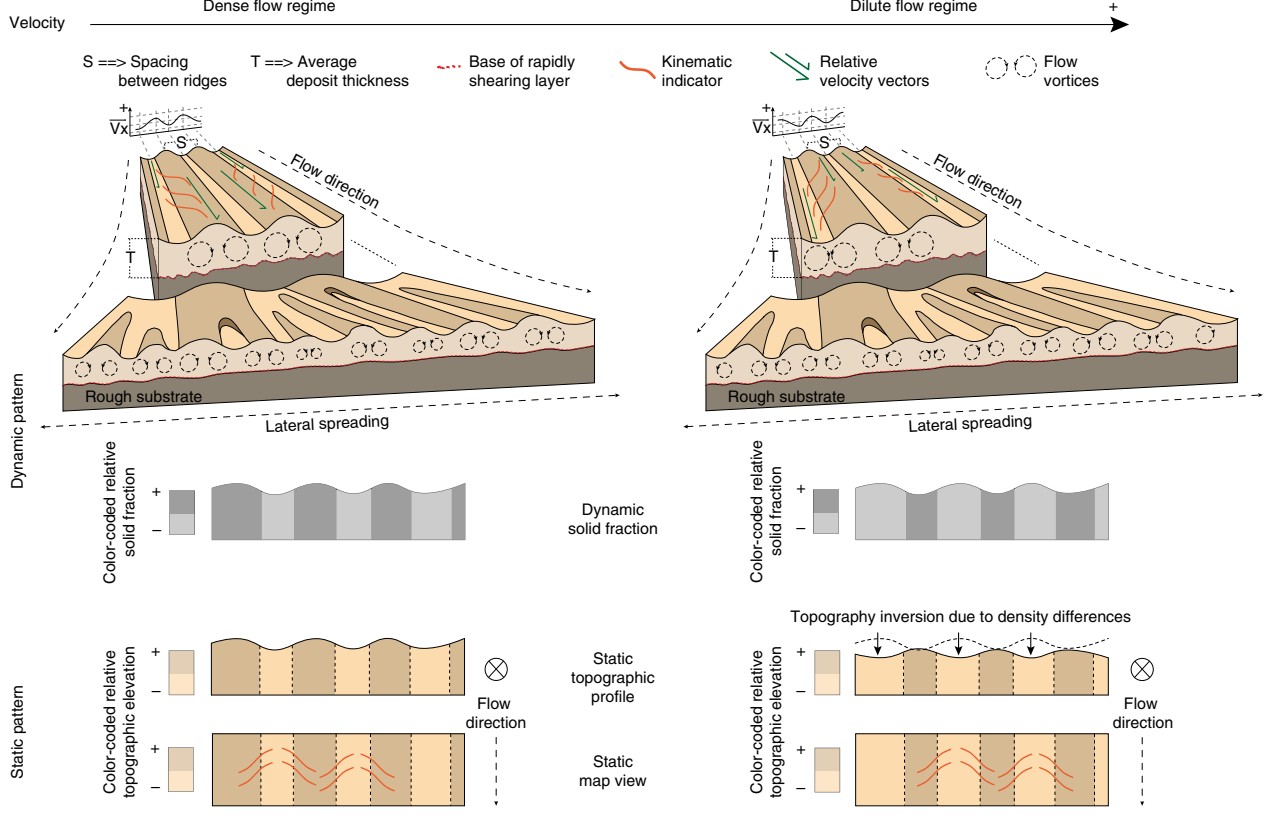

**Fig. 4** Flow structure models. Illustration of the two possible flow structure scenarios as inferred from the orientation of the en-echelon features observed superposing longitudinal ridges and interpreted as evidence of a velocity gradient between ridges and furrows. The current static pattern that we observe matches two dynamic patterns, a dense regime flow structure and a dilute flow regime structure followed by inversion of the topography as the flow stops and the ridges deflate

surface deposit of the martian landslide, $d = 1$–20 m (Supplementary Fig. 4), using HiRISE images (resolution 25 cm/px). We infer that the velocity of the martian landslide could range between 77 m/s and 345 m/s. Note that our estimation of the grain size may be biased towards higher values by resolution limitation and by a possible grain size segregation effect that brings larger grains closer to the surface. As such, we are aware that the range provided may exaggerate the most effective grain size and, therefore, the inferred velocity may be overestimated.

## Data availability
Imagery dataset used to generate DEMs and Orthophotos analysed in this study are available through the NASA PDS website http://pds-geosciences.wustl.edu/missions/mro/default.htm. We also provide the CTX and HiRISE image files and stereo-derived CTX DEMs and Orthophotos used in this study through data repository figshare https://doi.org/10.5522/04/9759266. All results generated during the morphometric analysis conducted in this study are included in this published article and its Supplementary Information.

## Code availability
Code used to generate some of the dataset analysed in this study is available from the corresponding author upon request.

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

## Acknowledgements
G.M. and T.M.M. acknowledge Science and Technology Facilities Council (STFC) funding ST/N504476/1.

## Author contributions
G.M. & T.M.M. came up with the concept. P.M.G., L.G. and H.H.S. contributed to the paper writing led by G.M.

## Competing interests
The authors declare no competing interests.
