## [Peer Review File · Nature Communications]

Reviewers' comments:

Reviewer #2 (Remarks to the Author):

This short paper uses the prominent longitudinal ridges visible on two lobes of a large landslide deposit in Valles Marineris, Mars to test several small-scale models of such ridge formation. While the authors of the present paper did not propose these models [ref. 16], they measured the ridge spacing on the Martian examples, and estimated the thickness of the landslide lobe to conclude that the principal control on the ridge spacing is the lobe thickness. They further argue from theory and laboratory experiments that a low-friction substrate, such as ice, is not required for ridge formation.

This is an interesting and important conclusion and certainly merits publication in Nature Communications. Such longitudinal ridges on landslide deposits have long intrigued many observers and this paper will serve to focus discussion on the meaning of these features. The paper also notes morphological evidence (S-shaped cross-ridges) that support the inference of differential velocities between the crests and troughs of the ridges—another important observation and conclusion.

I do have some small suggestions and concerns outlined below:

Introduction, end of paragraph 2, Ref. 19 is cited for longitudinal ridges on the Moon, however the longitudinal ridges in the Tsilokovsky ejecta have long been known and described by many authors, starting with Guest in 1971. A mention and citation would be appropriate here (Boyce et al, LPSC 47, Abstract #2471 would be OK, but perhaps a full publication would be better).

The authors cite a grain size of ca. 1-20 m for "grains" in the landslide, mainly derived from an image in Extended Data Figure 3. It is by no means clear what this size really represents. Because it is close to the limit of resolution of the images, this may merely be a largest grain size, rather than a representative size. My own experience with large landslide deposits indicates that the largest clasts usually work their way to the top of the deposit, with finer grained material below (this is especially evident in the deposit of the Frank, Alberta sturzstrom), in which case using the observed surface boulder size as a representative size may exaggerate the most effective grain size and, through their formulas for velocity, overestimate the inferred emplacement velocity. I would, in a revised discussion, like to see more acknowledgement that the slide lobe may contain a variety of grain sizes and that the surface boulder size may be a biased sample. While the theoretical and laboratory studies cited by the authors may have assumed a single grain size, it is most unlikely that a natural landslide would be so well sorted. Yet the readers of the present paper could easily get the impression that the authors believe that the slide exhibits only a narrow range of fragment sizes.

Methods section: While the authors do describe how the thickness of the landslide deposits was measured, they do not provide any estimates on how accurately they could reconstruct this parameter. Given that the relation between ridge spacing and lobe thickness is one of the principal conclusions of the paper, I would have liked to see an error estimate of this surface, as well as a plot of ridge spacing vs. slide thickness (thus illustrating the data reported in Extended Data Tables 2 and 3). As it is, the reconstructed valley floor topography is shown as green lines in Extended Data Figures 1 and 2, but no error estimation or uncertainties are indicated.

I have no objection to the authors knowing my identity.

H. J. Melosh

Reviewer #4 (Remarks to the Author):

The paper by Magnarini et al presents an analysis of the topography of a Martian landslide, which exhibits distinctive parallel ridges. The authors have developed image analysis to measure the typical wavelength of the pattern and they show that additional fine structures exist on top of the ridges, revealing some transverse velocities.

The wavelength of the longitudinal ridges is found to be between 2 and 3 times the thickness of the deposit. The authors discuss these observations in the light of previously published laboratory experiments, which have shown that such longitudinal pattern can be observed in rapid granular flows, due to an instability induced by the development of a strong basal shear and a dilute basal region.

I have found the paper interesting, suggesting an alternative and new explanation for the ridges observed in large landslides.

However, I think that the paper should be revised before publication, my comments being described in details below

1) The paper lacks some information about the measurements.

- First it would be interesting to mention in the main text and not only in the method, that the thickness is computed from the extrapolation of the basal topography from the measurement of the free surface outside the landslide. How precise is this estimate? especially for the profiles far from the edges (1 and 2 in Fig. 1) ?

-Secondly, It would be interesting to add in Fig. 1 a plot showing two or three thickness profiles (the red lines). This will provide in a glance information about the scale of the wavelength, about the thickness, about the amplitude of the deformation, and will also show the degree of periodicity of the ridges and the disorder of the pattern along the profiles.

- Thirdly, It would be important to show the correlation between the wavelength and the thickness in a plot, and not only to discuss it in the text. The authors should plot the wavelength versus the estimated thickness for all the points along the selected profiles. The reader would then have information about the range of wavelengths, about the range of thicknesses, about the dispersion of the data around a linear fit. Perhaps the authors can put in different colours the data corresponding to the central region and the one corresponding to the east area, which seems to show different linear correlation.

2) I think the authors should be more careful and more precise in their physical interpretation when discussing the comparison with the laboratory experiments. To my opinion they have made some mistakes in the interpretation.

In the lab, two kind of patterns have been observed. For very rapid flows, the surface deformation is sinusoidally deformed, and the transverse flow goes from the crest to the troughs, the troughs flowing faster than the crests and being much more dense (ref 14 and 16). For less rapid flows, a more complex pattern arises, with a secondary modulation and additional crests forming in the rapid regions (ref 16 , fig 2b).

However none of the pattern observed experimentally is the one sketched in Fig4 of the present paper, where the authors show a sinusoidal deformation with crests moving faster and attracting the lateral particles.

Another question not discussed by the authors concerns the possible difference between the pattern developed during the rapid flow and the pattern observed when the flow stops. From ref. 14, it has been shown by measurements of the light transmission, that the total mass of grains in the crests is less than the total mass of grains in the troughs. This means that if the flow stops suddenly, one expects the crests to collapse when flow ceases, implying that what was a trough during the flow becomes a ridge in the deposit. I am not aware of any experiments showing this, but the observation of the density variation suggests that the interpretation of the deposit analysed by the authors should be take with great care.

This point should be discussed in the paper.

3) The estimate of the velocity is to my opinion irrelevant. The velocity of the landslide can not be estimated by \sqrt{gd} . In steady granular flows, the mean velocity of a Bagnold profile scales like $\sqrt{g} h^{3/2}/d$ times a function of the inclination. But even this scaling is to my opinion no relevant for highly unsteady flow like landslides. The velocity certainly mostly depends on the initial height of the cliff and not the particle size.

In conclusion, I think that although the analysis of the Martian landslide is nice and interesting, although the connection with the longitudinal vortex instability is relevant and fruitful, I think the paper needs to be modified by taking into account the above comments before being published.

Reviewer #5 (Remarks to the Author):

This paper studies the origin of the parallel ridges often observed on the surface of gravity flows in planetary bodies. By reconstructing the 3D topography of the deposit of a Martian landslide, authors suggest that the ridge pattern results from a dynamic flow instability already reported at the scale of the laboratory with dry grains, rather than from the presence of a basal layer of ice as currently interpreted.

Being not an expert in planetary sciences or geomorphology, I have found the paper stimulating in that it links some morphology of large-scale planetary flows to granular flow instabilities observed in the laboratory, raising possible new mechanisms for their formation. However, for the paper to be fully convincing, several issues must be addressed.

1) My main comment is about the data processing and the morphological characterization methods. Authors write: "Distances between ridges were measured manually with the ruler tool. Number of ridges and their density along each profile were obtained using a custom Python script (See Code Availability)". Have authors tried to use a more robust and systematic method (Fourier transformation) to identify a typical wavelength? How did the authors manually identify a ridge? I ask these questions because, based on the raw profiles given in the Extended Data Figures, it is not clear how the data provided in Extended Tables 2 and 3 are extracted. As most of the paper is based on these data, wavelength extraction should be better explained and confronted with a non-manual and more objective measurement. The density must also be precisely defined.

2) My second comment is about the instability mechanism and the flow pattern shown in Figure 4. Authors should be cautious when interpreting the flow structure from the deposit pattern, which is static. The static deposit reported by the authors actually matches both the "dilute" flow structure proposed by Forterre and Pouliquen (as sketched in Fig. 2c Borzsonyi et al, PRL 2009) and the "dense" flow structure sketched in Figure 4. In the "dilute" flow structure, the troughs flow faster than the crests but are much denser, such that the integral of mass across a trough is higher than across a crest (Fig. 4 Forterre and Pouliquen PRL 2001). As a result, when the flow stops, there is an inversion: the troughs of the flow become static ridges while the crests of the flow become static furrows, as in the Martian deposit. I therefore suggest that both mechanisms (the dilute and dense) be sketched in Figure 4, since the deposit alone (thickness modulation and en-echelon structures) cannot discriminate between them.

3) My last comment is more on the form but important as well. I think Figure 2 is not very informative in the present version while the most important information is given in Extended Table 2 and 3. I suggest that authors add a graph in Figure 2 plotting the local thickness as function of the local ridge spacing for all the measurements given in Ext. Table 2 and 3.

Response to reviewers

We thank the editor and reviewers for their extremely useful comments on the paper. We have now produced a revised version that we feel is substantially improved due to these useful comments. We address each of these below in turn, with our response to the reviewers in red. As requested, we highlight the changes derived from the useful comments using font-colour blue (in the main text and in the Extended Data). We added the document file 'Extended Methods' as a result of reviewers comments.

Reviewer #2 (Remarks to the Author):

This short paper uses the prominent longitudinal ridges visible on two lobes of a large landslide deposit in Valles Marineris, Mars to test several small-scale models of such ridge formation. While the authors of the present paper did not propose these models [ref. 16], they measured the ridge spacing on the Martian examples, and estimated the thickness of the landslide lobe to conclude that the principal control on the ridge spacing is the lobe thickness. They further argue from theory and laboratory experiments that a low-friction substrate, such as ice, is not required for ridge formation.

This is an interesting and important conclusion and certainly merits publication in Nature Communications. Such longitudinal ridges on landslide deposits have long intrigued many observers and this paper will serve to focus discussion on the meaning of these features. The paper also notes morphological evidence (S-shaped cross-ridges) that support the inference of differential velocities between the crests and troughs of the ridges—another important observation and conclusion.

We thank the reviewer for these overall very useful comments. We are happy to see that the reviewer views our work as an interesting and important conclusion, meriting publication in Nature Communications.

I do have some small suggestions and concerns outlined below:

Introduction, end of paragraph 2, Ref. 19 is cited for longitudinal ridges on the Moon, however the longitudinal ridges in the Tsiolkovsky ejecta have long been known and described by many authors, starting with Guest in 1971. A mention and citation would be appropriate here (Boyce et al, LPSC 47, Abstract #2471 would be OK, but perhaps a full publication would be better).

This is a good point, we have now modified the end of paragraph 2 (lines 48-50) and included two more references to this:

19. Guest, J. E. & Murray, J. B., Nature and origin of Tsiolkovsky crater, lunar farside. *Planet. Space Sci.* **17**, 121-141 (1969).

20. Boyce J. M., Mouginis-Mark P. & Robinson M., An LROC update: The Tsiolkovsky landslide. *LPSC XLVII*, 2471 (2016).

The authors cite a grain size of ca. 1-20 m for “grains” in the landslide, mainly derived from an image in Extended Data Figure 3. It is by no means clear what this size really represents. Because it is close to the limit of resolution of the images, this may merely be a largest grain size, rather than a representative size. My own experience with large landslide deposits indicates that the largest clasts usually work their way to the top of the deposit, with finer grained material below (this is especially evident in the deposit of the Frank, Alberta sturzstrom), in which case using the observed surface boulder size as a representative size may exaggerate the most effective grain size and, through their formulas for velocity, overestimate the inferred emplacement velocity. I would, in a revised discussion, like to see more acknowledgement that the slide lobe may contain a variety of grain sizes and that the surface boulder size may be a biased sample. While the theoretical and laboratory studies cited by the authors may have assumed a single grain size, it

is most unlikely that a natural landslide would be so well sorted. Yet the readers of the present paper could easily get the impression that the authors believe that the slide exhibits only a narrow range of fragment sizes.

This is a good point. As the reviewer states, larger blocks are known to move to the surface, which actually adds to the idea that there are vertical movements of material during landslide propagations. As we have moved the velocity calculation paragraph to the 'Extended Data', we have now added two sentences in such paragraph and modified the caption of Extended Data Figure 3 so to acknowledge that the slide lobe may contain a variety of grain sizes and that the surface boulder size may be a biased sample.

Methods section: While the authors do describe how the thickness of the landslide deposits was measured, they do not provide any estimates on how accurately they could reconstruct this parameter. Given that the relation between ridge spacing and lobe thickness is one of the principal conclusions of the paper, I would have liked to see an error estimate of this surface, as well as a plot of ridge spacing vs. slide thickness (thus illustrating the data reported in Extended Data Tables 2 and 3). As it is, the reconstructed valley floor topography is shown as green lines in Extended Data Figures 1 and 2, but no error estimation or uncertainties are indicated.

These are good points, and this is also a concern echoed by reviewer 2. As such, we have now included a thorough error analysis in our revised manuscript (See Extended Methods). Measuring the actual thickness is an issue that other papers have raised as related to the landslide volume calculation (e.g., McEwen, 1989; Harrison and Grimm, 2003; Quantin et al., 2004), however none of these works provide a quantitative error estimation. We adopted the method of interpolating the valley floor that surrounds the landslide deposit, as this is the method commonly used in the literature (e.g., Harrison and Grimm, 2003; Quantin et al. 2004; Conway and Balme, 2014). Nevertheless, we agree with Reviewer #2 that, given the importance of the thickness parameter to the principal conclusion of the paper, error estimation is of great significance in this case. The following is included in the Extended Methods:

We traced profiles transversal and longitudinal to the valley floor, adjacent to the central and east areas of study (Extended Methods Fig.1). This set of transects (yellow lines), in the specific their linear fits (burgundy lines), is compared to a mirror set of transects traced in correspondence of the landslide deposit (blue lines) so as to: 1) assess the plausibility of the overall elevations of our reconstructed bottom surface, and 2) to provide an error estimation of the average thickness that we provide in the manuscript.

- 1) As the floor of Valles Marineris gently dips towards east (in the figure, from left to right), the higher elevations of the blue lines 1 and 2 are plausible. However, we are also expecting a random natural roughness of the valley floor at various scales so that also blue lines 3 and 4 appear plausible, as their maximum offset (blue line 3) is about 80 m, which is within a reasonable magnitude of topographic variation.

2) As the parameter used to arrive to the principal conclusion of the paper is the average thickness, we decided to express the error as standard deviation of the elevation points of topographic profiles. To do so, we use the topographic profiles that we traced in the valley floor, adjacent to the central and east areas of study (yellow lines). As the topographic profiles that we traced for the morphometric analysis are parallel to neither transversal nor longitudinal direction, we determine the final error taking into account the propagation of error along these two axes (Equation 1), that is combining standard deviation for yellow lines 1 and 2 to obtain the error in the central area of study (c) and yellow lines 3 and 4 for the error in the east area of study (d), represented as error bars in Fig. 2b in the revised manuscript:

$$\sigma_k = \sqrt{\sigma_i^2 + \sigma_j^2} \quad (\text{Equation 1})$$

Profiles	Standard deviation σ	Propagation of error σ_k
Yellow line 1	18.98 m (σ_i)	24.61 m (Central study area)
Yellow line 2	15.67 m (σ_j)	
Yellow line 3	25.86 m (σ_i)	67.9 m (East study area)
Yellow line 4	62.78 m (σ_j)	

The following figure is Figure 2b in the revised manuscript. As suggested by reviewer #2, we plot the average spacing of the ridges against the average thickness of the profile and adding the error bars. We added a grey band that shows the range within which the scaling relationship between the wavelength of the ridges and the thickness of the deposit is defined as in ref.14.

Reviewer #4 (Remarks to the Author):

I have found the paper interesting, suggesting an alternative and new explanation for the ridges observed in large landslides. However, I think that the paper should be revised before publication, my comments being described in details below.

We are happy to hear that the reviewer finds out paper interesting, we have found their comments very useful and have significantly improved the manuscript.

1) The paper lacks some information about the measurements.

- First it would be interesting to mention in the main text and not only in the method, that the thickness is computed from the extrapolation of the basal topography from the measurement of the free surface outside the landslide. How precise is this estimate? especially for the profiles far from the edges (1 and 2 in Fig. 1) ?

This is a good point, and is similar to the point raised by reviewer #2. We believe we have answered this in the revisions and response above.

-Secondly, It would be interesting to add in Fig. 1 a plot showing two or three thickness profiles (the red lines). This will provide in a glance information about the scale of the wavelength, about the thickness, about the amplitude of the deformation, and will also show the degree of periodicity of the ridges and the disorder of the pattern along the profiles.

We agree, and have now added the topographic profiles (f) of both the central study area (c) and the east area of study (d) to Figure 1, which makes it much easier for reader to understand the entirety of the ridge pattern along the profiles.

- Thirdly, It would be important to show the correlation between the wavelength and the thickness in a plot, and not only to discuss it in the text. The authors should plot the wavelength versus the estimated thickness for all the points along the selected profiles. The reader would then have information about the range of wavelengths, about the range of thicknesses, about the dispersion of the data around a linear fit. Perhaps the authors can put in different colours the data corresponding to the central region and the one corresponding to the east area, which seems to show different linear correlation.

This is similar to reviewer #2, and we have now modified Figure 2: Figure 2a gathers the boxplots for both the areas of studies that were previously separated; Figure 2b shows the average spacing of the ridges against the average thickness of the profile and the error bars.

2) I think the authors should be more careful and more precise in their physical interpretation when discussing the comparison with the laboratory experiments. To my opinion they have made some mistakes in the interpretation.

In the lab, two kind of patterns have been observed. For very rapid flows, the surface deformation is sinusoidally deformed, and the transverse flow goes from the crest to the troughs, the troughs flowing faster than the crests and being much more dense (ref 14 and 16). For less rapid flows, a more complex pattern arises, with a secondary modulation and additional crests forming in the rapid regions (ref 16 , fig 2b).

However none of the pattern observed experimentally is the one sketched in Fig4 of the present paper, where the authors show a sinusoidal deformation with crests moving faster and attracting the lateral particles.

Part of this point is addressed in the next paragraph, in which we show our modified model. Regarding the point about the secondary modulation and additional crests formation in the case of less rapid flows: first of all, if indeed this complex pattern arises we would expect having mapped the additional crests as ridges and so not discriminated as secondary modulation; secondly, as similarly discussed in Reviewer #5 about specific wavelengths and Fourier analysis, we know that ridges diverge with the distance and that at some point new ridges appear in between the diverging ones (Fig. 3a, 3b, 3c, 3d); therefore, we possibly cannot expect to distinguish secondary modulation from new forming ridges in such a constantly transforming pattern.

Another question not discussed by the authors concerns the possible difference between the pattern developed during the rapid flow and the pattern observed when the flow stops. From ref. 14, it has been shown by measurements of the

light transmission, that the total mass of grains in the crests is less than the total mass of grains in the troughs. This means that if the flow stops suddenly, one expects the crests to collapse when flow ceases, implying that what was a trough during the flow becomes a ridge in the deposit. I am not aware of any experiments showing this, but the observation of the density variation suggests that the interpretation of the deposit analysed by the authors should be taken with great care.

This point should be discussed in the paper.

This is a really good point. We have now modified the paragraph in the revised manuscript where we discuss the en-echelon as kinematic indicators (lines 120-130 and re-sketched our model in Figure 4 and changed the caption in the revised manuscript.

3) The estimate of the velocity is to my opinion irrelevant. The velocity of the landslide can not be estimated by \sqrt{gd} . In steady granular flows, the mean velocity of a Bagnold profile scales like $\sqrt{g} h^3/2/d$ times a function of the inclination. But even this scaling is to my opinion no relevant for highly unsteady flow like landslides. The velocity certainly mostly depends on the initial height of the cliff and not the particle size.

We are sorry, but we do not to agree with reviewer #4 on this point. We estimate the velocity that the landslide supposedly had in order to develop the longitudinal ridges. We apply the formula as in ref.16 and we use appropriate values, as given in ref. 16, for which the longitudinal pattern develops. The range of velocities we provide, although may

be overestimated as explained following a comment of reviewer #2, is consistent with other works on long runout landslides, in which high velocities have been estimated using other methods (e.g., Goren et al., 2010; Roberts and Evans, 2013; Mazzanti et al., 2016). This is indeed relevant and important information as it strengthens the evidence that long runout landslides are fast and able to develop (or maintain) high velocities even along the nearly horizontal part of their path. This behavior still lacks of a satisfactory explanation and therefore we consider this information worthy of inclusion. We have now moved the velocity estimation in the Extended Data.

In conclusion, I think that although the analysis of the Martian landslide is nice and interesting, although the connection with the longitudinal vortex instability is relevant and fruitful, I think the paper needs to be modified by taking into account the above comments before being published.

Reviewer #5 (Remarks to the Author):

Being not an expert in planetary sciences or geomorphology, I have found the paper stimulating in that it links some morphology of large-scale planetary flows to granular flow instabilities observed in the laboratory, raising possible new mechanisms for their formation. However, for the paper to be fully convincing, several issues must be addressed.

We are happy to see that the reviewer #5 finds the paper stimulating, and that we have raised a possible new mechanisms for the formation of the longitudinal ridges associated with long runout landslides. These comments are very useful, and we have addressed below.

1) My main comment is about the data processing and the morphological characterization methods. Authors write: “Distances between ridges were measured manually with the ruler tool. Number of ridges and their density along each profile were obtained using a custom Python script (See Code Availability)”. Have authors tried to use a more robust and systematic method (Fourier transformation) to identify a typical wavelength? How did the authors manually identify a ridge? I ask these questions because, based on the raw profiles given in the Extended Data Figures, it is not clear how the data provided in Extended Tables 2 and 3 are extracted. As most of the paper is based on these data, wavelength extraction should be better explained and confronted with a non-manual and more objective measurement.

We first address the question about identification of a ridge. A ridge is determined by its horizontal continuity in the direction of flow and its presence is obvious from a top-view observation of the landslide deposit, in the same way as ref.14 clearly observed what they call “a streaked pattern”. We mapped the ridges as they appear as an unmistakable morphological feature.

We agree in part, that Fourier transform helps pick out characteristic wavelengths. However, we did run Fourier analysis for each profile (see following figures; blue dotted lines are our average wavelength values as used in the manuscript), from which a typical wavelength does not stand out. Here are our reasons to think that, nevertheless, it does not compromise our results: 1) The geomorphological signal usually carries noise that derives from other superficial features and weathering processes, masking the target feature, making not trivial its distinction. In our case, this some-hundreds-million-years old martian long runout landslide has been subject to impact cratering, wind erosion, mega ripples formation in topographic lows, and so on, which inevitably add noise to the topographic signal. 2) We are not expecting that these field-scale morphological features would have a signal as clear as one obtained at laboratory –scale under controlled experimental conditions, for instance, the roughness; ref.14 glued one layer of particles onto a glass plate that covered about the 70% of the surface; given the diameter of the particles, 0.25 mm, and the thickness of the flow, about 2/2.5 mm, the artificial roughness represents a small fraction of the flow thickness; in our field-scale martian slide (as seen in the figure where we discuss the thickness error estimation for reviewer #2), we can expect topographic oscillations that range from few meters to 60/80 meters, which, given that the slide thickness decreases from about 400/500 meter in proximal areas to about 60/70 meter at the terminal edge, represent a larger fraction compared to the

laboratory case; we expect these topographic variations to have some degree of influence on the surface of the slide and so to interfere with a “perfect” pattern. 3) Finally, and probably more importantly, we know that ridges diverge with the distance and that at some point new ridges appear in between the diverging ones (Fig. 3a, 3b, 3c, 3d); therefore, we possibly cannot expect to obtain a typical wavelength, as these are, during the motion, changing and transforming features, although producing an unmistakable pattern.

Central area - Profile 1

Central area - Profile 4

Central area - Profile 2

Central area - Profile 5

Central area - Profile 3

Central area - Profile 6

Central area - Profile 1

Central area - Profile 5

Central area - Profile 2

Central area - Profile 6

Central area - Profile 3

Central area - Profile 7

Central area - Profile 4

The density must also be precisely defined.

Agree, we have now specified the unit of measure m^{-1} (number of ridges/length of profile) in the text (line 87) and in Extended Data Table 2 and 3.

2) My second comment is about the instability mechanism and the flow pattern shown in Figure 4. Authors should be cautious when interpreting the flow structure from the deposit pattern, which is static. The static deposit reported by the authors actually matches both the “dilute” flow structure proposed by Forterre and Pouliquen (as sketched in Fig. 2c Borzsonyi et al, PRL 2009) and the “dense” flow structure sketched in Figure 4. In the “dilute” flow structure, the troughs flows faster than the crests but are much denser, such that the integral of mass across a trough is higher than across a crest (Fig. 4 Forterre and Pouliquen PRL 2001). As a result, when the flow stops, there is an inversion: the troughs of the flow become static ridges while the crests of the flow become static furrows, as in the Martian deposit. I therefore suggest that both mechanisms (the dilute and dense) be sketched in Figure 4, since the deposit alone (thickness modulation and en-echelon structures) cannot discriminate between them.

This is a really good point. As this point was also raised by Reviewer #4, we think we have addressed it above.

3) My last comment is more on the form but important as well. I think Figure 2 is not very informative in the present version while the most important information is given in Extended Table 2 and 3. I suggest that authors add a graph in Figure 2 plotting the local thickness as function of the local ridge spacing for all the measurements given in Ext. Table 2 and 3.

As this is a point raised also by reviewer #2 and #4, we think we have addressed this above.

REVIEWERS' COMMENTS:

Reviewer #2 (Remarks to the Author):

The authors have done a commendable job of responding to my and the other referees' comments. The revised paper is a substantial improvement over the original and I now support publication of the revised MS as it stands.

H. J. Melosh

Reviewer #4 (Remarks to the Author):

The authors have answered to my comments and have modified the manuscript. I think the paper is stimulating both geomorphology and physics community and I recommend the paper for publication in nature Communication.